# Peer-to-Peer System Design Trade-Offs: A Framework Exploring the Balance between Blockchain and IPFS

Ámbar Tenorio-Fornés [1], Samer Hassan [1,2,*] and Juan Pavón [1]

1   GRASIA, Institute of Knowledge Technology, Universidad Complutense de Madrid, Decentralized Science, Decentralized Academy SL, 28040 Madrid, Spain; atenorio@ucm.es (Á.T.-F.); jpavon@fdi.ucm.es (J.P.)
2   Berkman Klein Center for Internet and Society, Harvard University, Cambridge, MA 02138, USA
*   Correspondence: shassan@cyber.harvard.edu

**Abstract:** The current state of the web, which is dominated by centralized cloud services, raises several concerns regarding different aspects such as governance, privacy, surveillance, and security. A way to address these issues is to decentralize the platforms by adopting new distributed technologies, such as IPFS and Blockchain, which follow a full peer-to-peer model. This work proposes a set of guidelines to design decentralized systems, taking the different trade-offs these technologies face with regard to their consistency requirements into consideration. These guidelines are then illustrated with the design of a decentralized questions and answers system. This system serves to illustrate a framework to create decentralized services and applications that uses IPFS and Blockchain technologies and incorporates the discussion and guidelines of the paper, providing solutions for data access, data provenance, and data discovery. Thus, this work proposes a framework to assist in the design of new decentralized systems, proposing a set of guidelines to choose the appropriate technologies depending on the relevant requirements; e.g., considering if Blockchain technology may be required or IPFS might be sufficient.

**Keywords:** decentralization; distributed systems; P2P systems; IPFS; Blockchain; multi-agent systems

## 1. Introduction

Centralized cloud web services now represent an increasingly large portion of the internet [1]. This trend has been significantly accelerated since the emergence of the Web 2.0 model [2], in which web applications enable user participation and user-generated content. Thus, today's internet activity is concentrated on highly successful web services which have dominance over their respective markets [3,4]. During recent years, concerns have been increasing on the multiple issues caused by this situation, with respect to, e.g., privacy [5], governance [3,6], legislation [1], surveillance [7], or security [8]. Consequently, there have been several proposals to tackle some of these issues through new legislation [9,10] or through recommendations for platform developers [11]. In parallel, these issues have triggered the emergence of a wide range of technical solutions through different forms of decentralization.

We may divide the proposed decentralized solutions into three waves. The first wave involved the use of "federated" technology [12–14]—i.e., multiple central nodes communicating with each other—where users are free to choose the node with which to interact . Email is a classic example of an open protocol that is federated, together with the more recent XMPP for chatting [15], OStatus for microblogging [16], ActivityPub for social networking [17], OAuth for authentication [18], or SwellRT for real-time collaboration [19]. This approach is based on interoperability across services and servers [12,20,21]. However, many of these technologies are still hindered by several drawbacks, such as the existence of points of failure [22] and control [23], or the lack of interoperability of the data beyond a few applications [14,21].

The second wave of decentralized solutions was achieved through fully distributed technology; i.e., P2P networks without classical servers but instead using ordinary computers (different from classical cluster/grid parallel computing). There have been multiple attempts to offer P2P web services [24,25], such as Freenet for censorship-resistant communication [26], although broad adoption has mostly been limited to the field of file sharing; e.g., eDonkey, BitTorrent [27].

The third wave began when some unresolved technical challenges with P2P solutions [28,29] became more evident. This opened the door to a new generation of solutions, most of which rely on cryptographic hashes organized in Merkle trees [30]. The advent of the first fully decentralized digital currency, Bitcoin [31], triggered a plethora of decentralized solutions based on its underlying technology: the Blockchain. In addition, another groundbreaking technology emerged around P2P storage: the IPFS, or Inter-Planetary File System [32]. These two new decentralized technologies, often combined, enable a wide range of applications [33–37]. Furthermore, CRDT (refer to Abbreviations for a list of acronyms and their meanings) [38] technology enabled real-time collaboration for P2P systems.

Exploring the synergies of these technologies may unveil new decentralization possibilities. IPFS is frequently used as a decentralized storage for Blockchain applications. However, other non trivial combinations of these technologies may enable new, decentralized system designs.

Therefore, there is a need for frameworks and models to explore the limitations and synergies of these recent innovations. This work proposes a combination of IPFS and Blockchain technologies for the design and implementation of open distributed systems. Concretely, it presents the trade-offs that decentralized technologies face and proposes design guidelines to assess the adequacy of the different considered technologies. Moreover, this paper attempts to assist computer scientists and software engineers in the design of novel distributed systems, proposing a set of guidelines to choose appropriate technologies depending on the relevant requirements. For instance, this framework would enable researchers to decide if Blockchain technology is really needed or if other alternative peer-to-peer technologies (such as IPFS) may be sufficient for the given use case.

The rest of the paper is structured as follows. Section 3 defines characteristics of the considered distributed systems. Then, Section 2 introduces the used decentralization technologies. Section 4 discusses the trade-offs of open distributed system design, discusses the tensions and approaches for consistency in such systems, and provides design guidelines to assess whether a system may require the use of Blockchain technology. Afterwards, Section 6 applies the previous section's discussions and design guidelines to propose a distributed system design, using a distributed questions and answers (Q&A) system as an example. The conclusions follow in Section 7.

## 2. Decentralization Technologies

Our proposal relies on Blockchain [31] and IPFS [32] decentralization technologies. This section describes these technologies and some of their underlying concepts and properties, such as content-addressability and Merkle linked structures.

**Content Addressability:** In centralized and federated systems, content is frequently referenced with addresses that include location information: Uniform Resource Locators (URLs) [39]. However, references to content can also be independent from their location, using Universal Resource Identifiers (URIs) [40]. In peer-to-peer systems, agents cannot rely on the location of other agents to access content because the content could be provided by any agent. The hash of any content can be used as its URI (hash functions are one-way collision-free functions; i.e., functions that result in a negligible probability of guessing which input produced an output). Thus, these hash URIs are used in multiple distributed systems such as IPFS to build scalable content-addressable networks [32,41–43].

**Merkle Links and Structures:** The use of hash values (see previous subsection) to reference data in data structures was first introduced in 1987 by Merkle [30]. Complex data structures can use these links (See Figure 1 for an example). Merkle-linked structures are key to the building of technologies such as Git [44], Blockchain [31], and IPFS [32], among others. Section 6.2 proposes the use of these structures for the data representation of the system.

**Blockchain:** Blockchain was the first technology that enabled a fully distributed digital currency (Bitcoin) [31], solving the double-spending problem in distributed systems (see Figure 2). It uses a Merkle-linked list of blocks of transactions (a Blockchain) to build a distributed ledger of transactions. To address the double-spending problem, it made it computationally difficult to propose a candidate for the next block in the distributed ledger and incentivized nodes to try to propose those blocks with valid transactions. Then, the protocol considers the largest observed chain the actual ledger to trust. Therefore, in order to forge a Blockchain, an actor would need half of the computing power of the system, bringing security to the consistency of the data recorded in the ledger. Section 4.4 proposes the use of the Blockchain to provide consistency to open distributed systems.

**IPFS:** Some peer-to-peer systems such as P2P sharing software [41] use a hash of the content to address it (See Figure 3). Other technologies such as Git use complex Merkle-linked structures [44]. IPFS integrates both the use of complex Merkle-linked structures with the data-addressability of P2P file-sharing systems. The content is distributed over a peer-to-peer network. Section 6.1 proposes the use of IPFS for the storage and distribution of data in the framework.

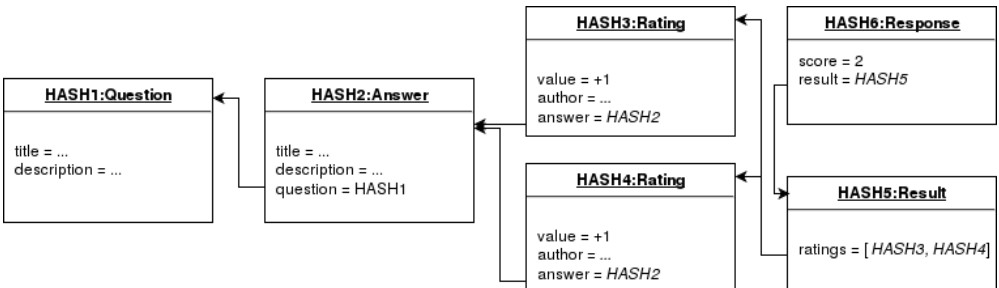

**Figure 1.** Merkle-linked data of an example Q&A system (such as Stack Overflow).

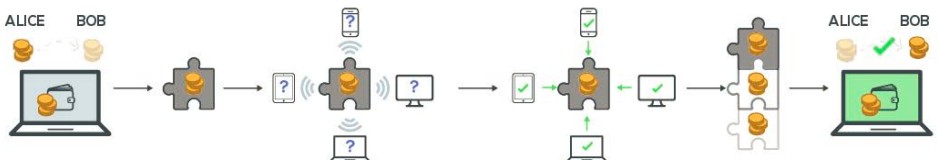

**Figure 2.** A brief Bitcoin overview: when Alice sends Bitcoin to Bob, her transaction is represented by a block, which is broadcasted to the network. When it is validated, the block is attached to the Blockchain, the transaction is performed, and Bob gets the Bitcoin.

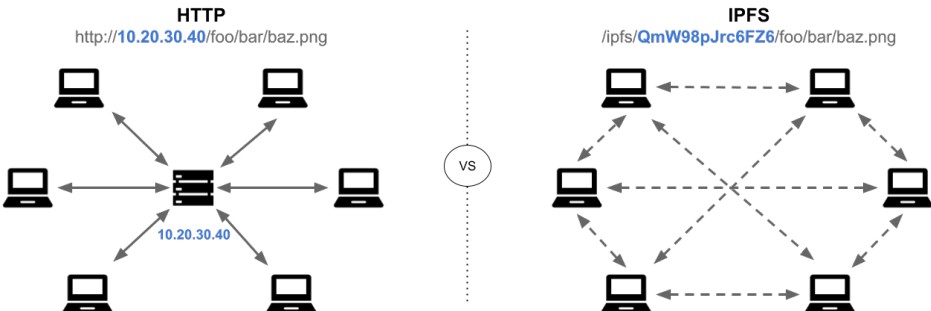

**Figure 3.** A brief IPFS overview: the classical HTTP protocol used in the web uses location addresses, relying on a centralized architecture in which users connect to the central server (location) which provides the file. Instead, IPFS uses content addresses, where users can retrieve files that are uniquely identified from any node in a distributed network that stores that file.

## 3. The Open Distributed Systems Considered

The main purpose of this work is to provide a framework and set of guidelines that may facilitate the design of open peer-to-peer services and applications that maintain a shared state and common agreed rules. This work focuses on *open* and *fully distributed peer to peer systems*, especially those enabled by the recent third wave of peer-to-peer solutions such as the Blockchain and IPFS (see Section 1). The kind of systems that are best suited for the given guidelines are defined in this section.

### 3.1. Shared State and Agreed Rules

Blockchain technology enabled a new generation of distributed systems. For the first time, distributed systems were able to maintain a consistent shared state and trust that the rules to change this state were strictly followed. This may be easily understood when considering the first application of the Blockchain: cryptocurrencies. In this context, there is a need for a consistent shared state: the amount of currency owned by each account. Similarly, nobody should be able to spend the same coin twice, and nobody should obtain money without either receiving it from a transaction or "mining" it for their contribution to the maintenance of the infrastructure.

To achieve this consistent shared state and strict rules, the Blockchain relies on the following building blocks:

- Agreed rules: A consensus on the rules of the system across the network. Thus, the participants on the network agree on how the shared state can change and who can change it. Cryptographic identities are used to ensure that the people who are performing the operations are allowed to do so. For instance, they may agree that only the owner of a cryptocurrency account can send money from that account.
- Trusted state: This is achieved using a tamper-proof, cryptographically-linked data structure named the Blockchain. Thus, every agent can access the complete history of transactions (the chain of blocks) and verify that the rules have been respected (e.g., that no one has sent more money than they initially had).
- Incentivized consistency: This ensures the maintenance required for the consistency of the shared state is performed appropriately. This is achieved by rewarding the "miners" for their maintenance work, typically through a Proof of Work or Proof of Stake algorithm.

This work studies the possibilities of new decentralized systems that rely on similar building blocks. We consider systems in which agreed rules can update the system state in a way that every agent can trust; i.e., it can be verified that the rules have been followed. This paper challenges two typical assumptions in Blockchain systems:

- The hard requirement of a single data structure (such as the blockchain) to maintain all the information is not suitable for multiple distributed systems. Thus, this paper

proposes that other distributed architectures such as an IPFS network may store and distribute such information.

- Similarly, maintaining consistency through an elaborate incentive system (such as the famous Proof of Work algorithm) is not a must for multiple distributed systems. Thus, the proposed design guidelines help users to assess whether a Blockchain is needed or if other consistency strategies may be followed (namely, consistency as logical monotonicity, or CRDTs).

Note that other kinds of distributed systems with purposes different than those defined (to maintain a shared state and agreed rules) fall outside the scope of this paper. Thus, classical distributed computing (as in grid computing) or Content Delivery Networks (CDNs) are not considered.

### 3.2. Openness

*Open systems should provide the means for autonomous agents to enter, interact, and leave the system.*

The concept of an open system has been widely applied in computing and telecommunications for a long time (see, for instance, standardization efforts such as the OSI model [45]). Its main idea is that services (with well-specified interfaces) can be provided by different entities with their own implementation. An open system, therefore, specifies the means for the communication of its entities, which can enter, interact, and leave the system [46,47].

The evolution of the open system is therefore highly dynamic, which makes it quite complex to obtain complete knowledge of the whole system state at any time. Entities only have a partial knowledge of their environment (the open system), and the only aspect that all entities hold in common is their ability to communicate with each other [47]. In this sense, the paradigm of multi-agent systems (MASs), which assume autonomy and the ability for distributed entities—the *agents*—to communicate to be fundamental, is a proper model for the development of open systems. An agent is an autonomous entity, with the assumption that its knowledge of the world is partial [48], so it tries to take the best decision (principle of rationality [49]) and interacts with other agents.

### 3.3. Peer-to-Peer Full Distribution

*Fully distributed peer-to-peer systems are composed of a network of interconnected agents that communicate and coordinate their actions without a central control entity.*

Systems such as the web and P2P file-sharing programs are distributed systems composed of web servers and computers sharing files, respectively [41,50]. While centralized systems depend on a single component for their operation, distributed systems are resilient to the disconnection of some of their components; e.g., if a web server is disconnected, the web will still be a functional system. However, some distributed systems still depend on single components for parts of the system to work. For instance, if a web server disconnects, their web pages will become unavailable. This work refers to *peer-to-peer systems* when referring to distributed systems that are independent from any single node.

## 4. Design Trade-Offs of Distributed Open Systems

The design of decentralized open systems faces some challenges. Unlike centralized systems, they lack a single entity to determine the consistency of the state of the system. This work focuses on the different strategies that decentralized systems can adopt to achieve consistency. Indeed, Blockchain technology was a solution for a specific problem (the design of a decentralized currency system) with a very strong consistency requirement: users should be able to know who owns money in the system and be sure that each transaction follows the agreed rules. However, not all distributed open systems have such strong consistency requirements.

Fortunately, the existing literature has extensively studied the issue of consistency in decentralized systems. This section builds upon some of the most relevant literature

on the consistency of distributed systems and provides a set of four guidelines to design distributed open systems.

First, Section 4.1 introduces the *CAP Theorem* [51], which provides a framework that states the unavoidable compromises between data consistency, availability, and partition resistance in distributed systems. Then, Section 4.2 explains the *CALM Principle*, which offers tools to discover if an open distributed system needs coordination technology for consistent behavior or if alternatively it can be achieved through *Logical Monotonicity*. Finally, Section 4.3 introduces Conflict-Free Replicated Data Types (*CRDTs*), which provide a solution to achieve *eventual consistency* for these systems without needing coordination technologies. Finally, Section 4.4 explains that using *Blockchain* enables such coordination technology to maintain consistency while preserving decentralization when CRDTs cannot be used or when the system has stronger consistency requirements.

### 4.1. CAP Theorem

The *CAP* Theorem [51] states that a networked data system can only hold two of these three desirable properties:

1. Consistency: The requests of the distributed system should behave as if they were handled by a single node with updated information.
2. Availability: Every request should be responded to.
3. Partition resistance: The system should be able to operate in the presence of network partitions.

Given that the framework considers open systems where agents with partial information can join or leave at any moment, partition resistance is a necessary property for our proposal. Therefore, one of the most important design decisions for the systems built within this framework is to find the best balance between consistency and availability.

### 4.2. CALM Principle

Some queries are impossible to resolve in distributed open systems. Intuitively, in a distributed open system, some data may not be accessible; therefore, queries that need to take into account all the information of the system such as those that count the data that satisfy some constraints (e.g., counting the exact number of web pages that include a certain word) are impossible to resolve.

The Consistency as Logical Monotonicity (CALM) principle describes those queries that can be resolved in a distributed system without coordination [52]. A system is considered to be *logically monotonic* if the truth of a given statement cannot change by considering new information. In such systems, the responses to distributed queries are consistent.

The designer of a distributed system can check the monotonicity of its queries as follows:

**Order independence:** This is a needed condition for logical monotonicity [52]; i.e., if the system behavior depends on the order in which the information is received, then it is non-monotonic. For instance, in the double-spending problem, where an agent tries to spend "the same coin" twice, the state depends on which payment was made first. Therefore, it is a non-monotonic problem.

**Monotonicity:** By definition, if new information may revoke a previously valid response to a query, the query is non-monotonic. For instance, counting the number of positive votes for an answer in a Q&A system is non-monotonic, since new votes would change the response.

**Formal analysis:** This can prove the logical monotonicity of a system [52].

In distributed open systems, non-monotonic queries may produce non-consistent results without a coordination mechanism. Thus, in the presence of non-monotonic queries, the designer should decide on the consistency requirements of the system.

**Guideline 1.** *Monotonic queries can be consistently resolved in open distributed systems without coordination technologies.*

Thus, in the presence of network partitions, choosing perfect consistency over availability can be implemented without coordination using logically monotonic systems. If inconsistent behavior, such as missing some votes in a Q&A system, is acceptable for the system, then coordination mechanisms are still not needed.

**Guideline 2.** *Consistency requirements are a design decision. If inconsistent behavior is acceptable for the non-monotonic queries of the system, coordination technologies are not required for open distributed systems.*

Moreover, some non-monotonic open distributed systems may achieve eventual consistency without coordination, as explored in the next subsection.

### 4.3. Eventual Consistency

Eventual consistency is defined as consistency among the nodes of a distributed system once all messages have been delivered. The proposed Conflict-Free Replicated Data Types (CRDTs) enable eventual consistency without coordination, such as reaching consensus or rolling back [38]. A data type is said to be a CRDT if the possible concurrent operations are commutative.

**Guideline 3.** *Eventual consistency can be achieved without coordination in open distributed systems by ensuring that concurrent operations are commutative.*

Note that with eventual consistency, statements that are considered true in a given time can become false after receiving new messages. Thus, this consistency may not be sufficient for systems with strong consistency requirements, such as crypto-currencies.

CRDTs achieve eventual consistency once all messages have been delivered. Different systems may tolerate different delays of these messages. For instance, while a Q&A system may ignore a vote for a long period of time, for a collaborative document, incorporating relatively old updates may be problematic, regardless of eventual consistency.

### 4.4. Blockchain for Distributed Consistency

Some non-monotonic problems, such as the double-spending problem in distributed currencies, require strong consistency. Thus, a coordination mechanism is needed to provide that consistency. Blockchain technology enabled the implementation of Bitcoin [31], the first distributed digital currency. It proposed a fully distributed coordination mechanism to establish a consensus on the order of valid transactions. Thus, it provided consistency to a non-monotonic problem in a fully decentralized system. Indeed, the Blockchain can be used to provide consistency to other non-monotonic systems by establishing a consensus on the order in which the information should be considered.

**Guideline 4.** *The non-monotonic queries of an open distributed system with strong consistency requirements should be supported by a coordination technology such as the Blockchain.*

The guidelines are summarized in Table 1.

**Table 1.** Guidelines summary.

|  | Weak Consistency | Eventual Consistency | Strong Consistency |
|---|---|---|---|
| **Weak availability** | No need for coordination technologies (Guideline 2) |  | Logical Monotonicity or Blockchain (Guidelines 1, 4) |
| **Strong availability** |  | CRDTs (Guideline 3) | *Not possible, considering CAP Theorem* |

## 5. Applying the Guidelines to Well-Known Decentralized Systems

To illustrate the use of the proposed guidelines, we briefly discuss how they would be applied to two well-known but very different decentralized systems: PGP [53] cryptographic key servers and the Git [44] version control system.

### 5.1. PGP Keyservers

Pretty Good Privacy (PGP) [53] is a de-facto standard for cryptographic identities used to secure email, providing cryptographic signatures and encryption. These identities are often uploaded and shared at PGP keyservers. These keyservers form a decentralized network in which users upload their key to one of the servers, but the servers need to update each other in order to keep the global set of public keys available for any user that requests keys from other users. Moreover, users validate (sign) each other's keys in a distributed manner, forming a "web of trust". Below, we describe how each of the guidelines applies to this system.

When Alice receives a signed message from Bob, she needs to verify that the signature indeed belongs to Bob. A first approach for this verification is to find a signed message from a person Alice trusts stating they trust that the key belongs to Bob. This is the basis of the web of trust for PGP identities. In this simplified first approach, this is a non-monotonic query, since if a signature from a trusted person says that the key belongs to Bob, no new information would override that statement. Following Guideline 1, if we do not need strong availability, we do not need coordination technologies. This is true for many uses of PGP, where users simply share their keys with the few relevant people that they use PGP to communicate with. Still, public keyservers have emerged for convenience, and in these systems, users could upload their public keys with the signatures of the keys they trust.

However, people soon realized that keys could be compromised or that people could revoke the trust they shared about a public key. Thus, the question of whether to trust a key can change in the face of new information, making the system non-monotonic. Facing this issue and following Guideline 2, the system designer should chose whether consistent behavior is acceptable or not (i.e. if it is required to stop trusting a key as soon as the owner or the trusted parties revoke their trust). Many PGP users believe this is a strong requirement and therefore use keyservers to share and update information about trusted keys. With respect to the operations of trusting and revoking trust of a key, Guideline 3 ensures that we will eventually achieve consistency (e.g., once the user receives meaningful revocations). However, trusting cryptographic keys is considered by many to be an issue with strong consistency requirements, as it is dangerous to trust a key that has been compromised. Thus, as Guideline 4 states, there is a need for coordination technologies. In the case of the PGP web of trust, this is achieved by a network of keyservers that update each other and that users can query whenever they need to verify the status of a public identity. Indeed, some have proposed the use of the Blockchain for this strong consistency requirement [54]. However, as we have discussed above when referring to Guideline 3, we are considering a problem in which meaningful consistency is eventually achieved. Thus, we might argue that the Blockchain is actually not needed, and a network of public keyservers that keeps updated information might be sufficient.

### 5.2. Git

Git [44] is distributed version control software that is widely used to facilitate collaboration and maintenance in software repositories. Similar to the Blockchain, users can verify that changes were made by authorized actors and see the whole history of the version of the repository to which they have access. Multiple different versions of a Git repository coexist on many different computers. Typically, there is an official version maintained by the official contributors, but each of these contributors might have changes that they have not shared yet, and many unofficial versions of the repository with different versions of the software might exist. One of the most common queries in Git is to know the current status of a specific version of the repository (this is a non-monotonic query, as new changes would change the response, as explained in Guideline 1).

Git is a fully decentralized system. Each user (Git instance) may query the changes performed by other users from other known instances in order to keep their information up to date. Thus, this semi-manual update process is the proposed "coordination" mechanism used to obtain an updated state of the system. However, and for convenience, many developers started to use a single repository for coordination purposes, and centralized services for that purpose, such as Github or Gitlab, gained popularity. To re-decentralize Git repositories and maintain the convenience of a single source of consistent information given by centralized services, some have proposed the use of the Blockchain [55]. Git obtains eventual consistency by design (Guideline 3), since it maintains a strict order of operations. Thus, the Blockchain is not needed as a coordination mechanism, since other systems to share updated information may be sufficient.

## 6. Designing a Distributed Questions and Answers System

In this section, the trade-offs and design guidelines introduced in this paper are presented through a running example of a simple Q&A system, such as the well-known Stack Overflow [56]. The balance between availability and consistency in the system is discussed, and the need for Blockchain technology is assessed.

The proposed system architecture relies on IPFS for fully distributed data storage, public-key identities for data provenance, and a peer-to-peer network for communication. This section introduces how data access, data provenance, and data discovery are provided by the proposal.

### 6.1. Accessing Data

In centralized Q&A systems such as Stack Overflow, data are addressed and accessed using a location-centric model; i.e., a server is responsible for providing the data. For instance, a user may search for responses to a programming problem on the Stack Overflow website.

The use of content-addressable models for data access provides a fully distributed alternative. Our architecture relies on the IPFS network to distribute the data as Merkle-linked structures. These data structures provide both a Merkle-linked structure and data addressability [32]. Concretely, the data in the system are composed of key–value records and by named, directed Merkle-links to other data (as depicted in Figure 1). This data may be provided by any agent of the system.

### 6.2. Data Provenance

In centralized and federated systems, the trustworthiness of the data is provided through a direct connection to trusted servers; e.g., the user of a centralized Q&A system trusts a server not to hide or alter the information of the system. Fully decentralized alternatives can also be considered to obtain trustworthy data.

We propose the use of asymmetric cryptography identities to ensure the trustworthy provenance of data. Data that are digitally signed by trusted identities are trusted in the system. Following the technological choices of the architecture, the use of IPNS [32] or the Ethereum [57] identity infrastructure can be used.

Following our Q&A system example, every question, answer, and vote is digitally signed by the authors. Replicating the behavior of Stack Overflow, every user can submit questions and answers to the system. Thus, every signed question or answer is consider valid. A simple version of the system may consider every question, answer, and vote valid, thus having weak consistency requirements. Such a system would not need coordination technologies (Guideline 2) to work. However, systems such as Stack Overflow implement strategies to avoid system abuses; for instance, the system only allows authors with at least 15 reputation points to vote. Five reputation points are earned with each positive vote to a question or answer. Thus, to implement such strategies, our system should only allow the votes of users with at least three positive votes. Since these votes also have to be valid, the vote verification is recursive, until it reaches a trusted base case; e.g., identities that were initially allowed to vote without reputation in the system.

If negative votes are not considered in the system, answering whether a vote is valid is a non-monotonic problem. Thus, it can be implemented in a distributed system with strong consistency without coordination mechanisms (Guideline 1). However, the recursive nature of the example shows that the size and complexity of the data needed to trust a response may not be trivial.

The consideration of negative votes to questions and answers that would decrease the reputation of the authors adds complexity to the problem. The question of whether an identity has at least 15 reputation points is no longer monotonic, since observing new negative votes may change the results. Fortunately, adding and subtracting values to a number are commutative operations. Thus, and following the proposal of CRDTs, we could chose availability over consistency and be able to operate in the system while not knowing all the up and down votes, trusting that eventual consistency will be achieved (Guideline 3).

Furthermore, digital signatures may not be enough to prove authorship in the system. A malicious agent may sign data previously authored by other agents. Deciding the identity of the first author is therefore a non-monotonic problem that cannot be resolved with strong consistency without coordination. This problem is similar to the double-spending problem and could be resolved using the Blockchain if the designer considers that the system requires such strong consistency (Authors could use blockchain to claim authorship by registering a fingerprint of their content in the blockchain before publishing it.) (Guideline 4).

Non-monotonic searches (see Section 4.2) with strong consistency requirements, such as determining the exact number of votes for a question, may need the use of the Blockchain as a coordination mechanism. For instance, the votes of a Q&A system or the authorship of questions and answers could be registered in a Blockchain to provide consistency to those queries. Our architecture proposes the development of smart contracts using Ethereum [57] to provide such consistency for these systems.

### 6.3. Data Discovery Using a Trustless Distributed Protocol

To discover data in our open and distributed system, we propose the use of a query protocol. The queries of the system state the constraint that the responses must satisfy. For instance, a question that contains a given text can be searched in a Q&A system. The query can also constrain the structure of the response (e.g., it has more than one answer and more than one positive vote).

Additionally, a score function can be defined to sort the valid responses. For instance, the questions containing some text can be ranked by the number of positive votes.

Next, the protocol interactions (Figure 4) are described as follows:

1. An agent sends a query (with constraints and a score function).
2. Any agent can reply, with a response consisting of a content-centric link to the data satisfying the query and its corresponding score.
3. The querying agent accesses the data and verifies the responses and scores.

This protocol presents the following characteristics:

1.  Lightweight communication: Responses consist of a short link and a numeric value. Their length is then a few bytes long, even though they may represent complex, large data structures.
2.  Early distributed ranking: Responses may be ranked without accessing their data.
3.  Trustless ranking and validity: Similarly to how any node can verify that all the transactions in a Blockchain are valid without needing to trust a third party, the validity and ranking of the responses in the proposed protocol can be assessed without trusting the agents providing the responses or the data (e.g., checking the digital signatures for authorship and the validity of the linked votes).

The protocol can be implemented using the following: (1) Merkle-linked data distributed over IPFS; (2) JavaScript pure functions to express query constraints and score functions, using the JavaScript implementation of IPFS; and (3) a bus model for distributed systems communication [58] over IPFS pub–sub channels. Thus, the protocol would enable the implementation of distributed open systems with different consistency and availability requirements (see Table 1 for a summary of the guidelines for those different requirements).

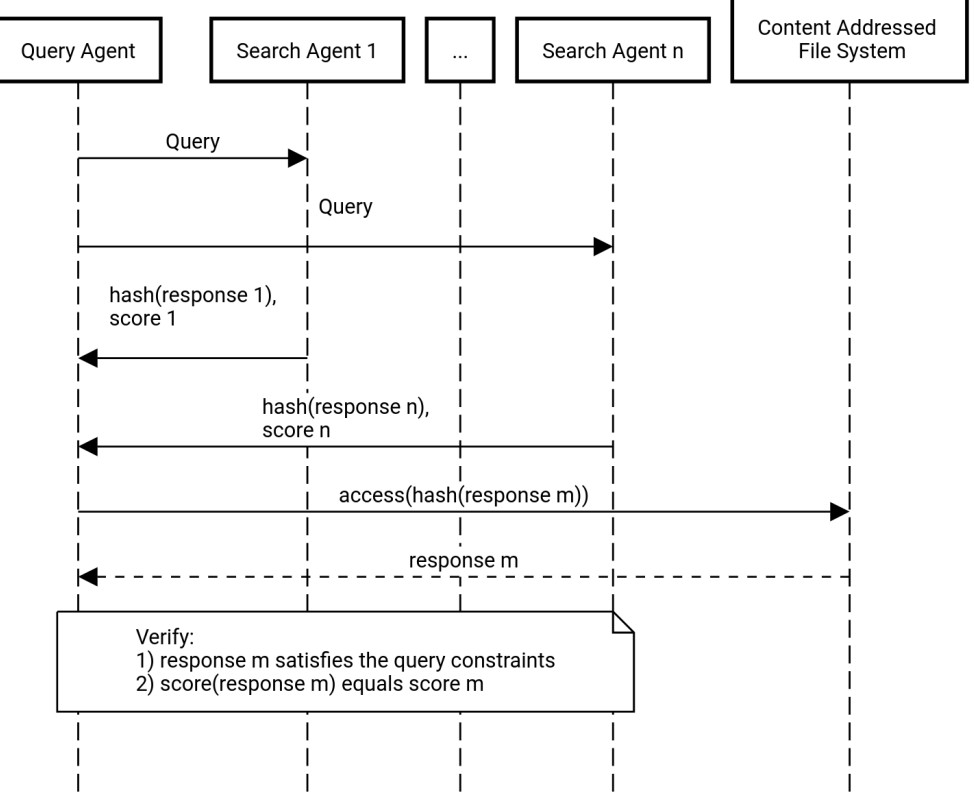

**Figure 4.** Distributed discovery protocol UML sequence diagram.

## 7. Discussion and Conclusions

This work introduces the tensions between consistency, availability, and partition resistance in fully distributed systems using current technologies such as the Blockchain and IPFS. It explores the possibilities and limitations of different approaches and technologies, providing guidelines to design these fully distributed systems. The guidelines help to assess whether blockchain technology may be needed for a distributed system. Four guidelines provide alternatives depending on the consistency and availability requirements of the system. The paper claims that these consistency and availability requirements are design decisions and that some systems may not have strong requirements for either of them, thus removing the need for advanced technologies to enhance coordination or availability (Guideline 2). For solutions that require strong consistency, logical monotonic systems can provide such consistency without coordination (Guideline 1). However, not

all problems are non-monotonic, and in that case, a Blockchain is required to provide such consistency and maintain the system decentralization (Guildeline 4). For systems with weaker consistency requirements, CRDTs offer an alternative that favor high availability while relaxing their consistency requirements to eventual consistency (Guideline 3).

The paper presents an architecture that is illustrated with a running example of a Q&A system. In this proposal, the data are represented as Merkle-linked structures and distributed with IPFS. Asymmetric cryptography provides trust to the data provenance of the distributed system. Ethereum technology is proposed as the Blockchain-based coordination framework to support the non-monotonic strong consistency requirements that these systems may have. A query communication protocol enables the data discovery in the open distributed system, providing ranked responses and the trustless verification of responses.

This proposal faces some limitations and challenges, as with other Blockchain-based and distributed technologies, such as privacy [59,60], sustainability [61], and scalability [62]. Ultimately, these challenges, which are not covered by our guidelines, could determine which distributed systems are viable. Furthermore, the design of distributed systems following our proposal should consider security concerns faced by similar distributed systems such as *sybil attacks* [31] and *generation attacks* [63]. Still, the sustainability and privacy of decentralized technologies is often better than the centralized alternatives [20].

Future work would help to consolidate and validate the contributions of this paper. Studying the efficiency and performance of the system, the proposal and implementation of new applications, the identification of more suitable network topologies and protocols, or the use of specialized agents such as search agents for specific applications are some of the opportunities to explore.

Decentralization technologies offer an opportunity to solve some of the challenges of the current internet. This paper has introduced design guidelines and a framework to design and build these systems using the potentials of new decentralizing technologies.

**Author Contributions:** Conceptualization, Á.T.-F.; methodology, Á.T.-F.; formal analysis, Á.T.-F. and J.P.; investigation, Á.T.-F.; writing—original draft preparation, Á.T.-F.; writing—review and editing, S.H. and J.P.; supervision, S.H.; project administration, S.H.; funding acquisition, S.H. All authors have read and agreed to the published version of the manuscript.

**Funding:** This research was funded by the project P2P Models (https://p2pmodels.eu, accessed on 25 October 2021) fundedby the European Research Council ERC-2017-STG (grant no.: 759207), Decentralized Science (https://decentralized.science, accessed on 25 October 2021) funded by the European Union's Horizon 2020 research and innovation program within the framework of the LEDGER Project (grant agreement No82526) and Chain Community, funded by the Spanish Ministry of Science, Innovation and Universities (grant RTI2018-096820-A-100).

**Institutional Review Board Statement:** Not applicable.

**Informed Consent Statement:** Not applicable.

**Data Availability Statement:** Not applicable.

**Acknowledgments:** We are grateful to the editor and the reviewers for their constructive feedback on our manuscript. We would like to thank Elena Martínez, from the P2P Models team, for her valuable work designing the included Bitcoin figure. We would like to express our gratitude to the Wikimedia Commons repository for their open-licensed laptop and server icons used in the IPFS figure. Finally, we would like to thank Alexandra Elbakyan (Sci-Hub) for her contribution to making scientific knowledge available for everyone.

**Conflicts of Interest:** The authors declare no conflict of interest.

## Abbreviations

The following abbreviations are used in this manuscript:

| | |
|---|---|
| CALM | Consistency as Logical Monotonicity |
| CAP | Consistency, Availability, Partition resistance |
| CDN | Content Delivery Network |
| CRDT | Conflict-Free Replicated Data Types |
| IPFS | Inter-Planetary File System |
| IPNS | Peer-to-peer |
| MAS | Multi-Agent System |
| OAuth | Originally "Open Authorization", open standard for access delegation |
| OStatus | Originally "Open Status", open standard for federated microblogging |
| P2P | Peer-to-peer |
| PGP | Pretty Good Privacy |
| Q&A | Questions and answers |
| URI | Universal Resource Identifier |
| URL | Uniform Resource Allocator |
| XMPP | Extensible Messaging and Presence Protocol (originally, *Jabber*) |

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
