# Peer review of "Peer-to-Peer System Design Trade-Offs: A Framework Exploring the Balance between Blockchain and IPFS"

_applsci, doi:10.3390/app112110012_

Round 1
Reviewer 1 Report
In the manuscript, the authors propose a set of guidelines in designing a decentralized system, especially leveraging blockchain and IPFS. Some comments and suggestions are as follows:
- The submitted file is in image and cannot be searched.
- The scientific value is weak. We don't see new problem or new method. Only descriptive guidelines are presented.
- In terms of designing decentralized system, are blockchain and IPFS suitable for all kinds of systems? If not, when will be blockchain and IPFS be applicable?
Author Response
Attached you may find a document detailing our responses to each of the comments.
Kind Regards

Reviewer 2 Report
The present study presents a guidline for the design of distributed applications. In general, the design of the study, the completeness of the review of existing problems and the originality of the study allows me to recommend this request for publication, after clarifying a number of issues:
1. In the review, consider other technologies. For example, projects aimed at distributed computing using grid-computing networks (https://en.wikipedia.org/wiki/Volunteer_computing)
2. If the article attempts to make a broad overview of distributed technologies, it is not superfluous to mention such a widespread technology as CDN.
3. At the beginning of section 4, there is a lack of justification for using the CAP Theorem and CALM Principle proposed in 2011-2021.
4. Section 5 describes the process of applying this guidline to the Q&A support system. But to test the model for adequacy, more cases are needed. For example, briefly describe the results of guidline application to any open and well-known distributed project.
5. Section 6 mentions privacy and sustanability issues to consider. But it is necessary to mention a few words about the scaling of such systems.
Author Response

(The authors gave the same response as above.)

Reviewer 3 Report
This paper introduces the guidelines of designing a decentralized systems with considering the technical tradeoffs under different scenarios. An example using IPFS and Blockchain to build a Q&A system is discussed to illustrate the ideas. This topic will be interesting for readers and help them design the distributed system. This paper is well written, clearly structured and easy to read.
The first suggestion is that the section-5 can be enhanced with more details; the second suggestion is that as the blockchain has been widely applied to different domains, the authors can pick up multiple popular cases to summarize/evaluate/compare these systems and illustrate how these systems are designed and the tradeoffs are made. The section-5 sounds a little weak to me.
There are a few small grammar issues (e.g., Line-324). It will be great if the authors can do another round of proof-reading.
Author Response

(The authors gave the same response as above.)

Reviewer 4 Report
Good work
Author Response
Thanks a lot for your kind review

Reviewer 5 Report
The paper proposes a set of guidelines to innovate in the future of the web-services. Authors give the example of StackOverflow to be adapted to its own proposal.
The paper is of interest and in my opinion it is of enough quality to be published in the journal. Nevertheless I have some suggestions to improve the quality of the paper:
- To improve the transversal state of the art developed in sections 1, 2 and 3, I would suggest a table with acronyms and its definitions, such as CRDTs, IPFS, XMPP, OStatus, OAuth, CAP, CALM, IPNS ...
- Table of figure 1 should be resized, although its content is of crucial interest.
- Although authors are experts, and also some readers, I would suggest to give an example of the "logically monotonic" property in the CALM Principle, and an example of CRDT. This properties are widely used in the rest of the paper.
The paper is quite hard to follow, nevertheless the contents are of quality. I would suggest to include some figures to explain the actual decentralized technologies, such as blockchain and IPFS, at least in section 3.
I dont understand why the query protocol named in section 5.3 is trustless. Please explain it.
Typo:
- 234: without with
Author Response

(The authors gave the same response as above.)
